# Spontaneous Pneumomediastinum in a 16-Year-Old Patient with SARS-CoV-2 Infection: A North-East Romanian Case

**DOI:** 10.3390/children9111641

**Published:** 2022-10-27

**Authors:** Florin Filip, Monica Terteliu Baitan, Olga Adriana Caliman Sturdza, Roxana Gheorghita Puscaselu, Roxana Filip

**Affiliations:** 1Faculty of Medicine and Biological Sciences, Stefan cel Mare University of Suceava, 720229 Suceava, Romania; 2Suceava Emergency County Hospital, 720224 Suceava, Romania

**Keywords:** spontaneous pneumomediastinum, COVID-19 infection, child, chest CT scan

## Abstract

Spontaneous pneumomediastinum (SPM) associated with SARS-CoV-2 infection is a rare condition but can represent a medical emergency. It is probably related to alveolar damage secondary to SARS-CoV-2 infection, which allows air to escape in the surrounding lung tissue. Cough and airways’ barotrauma are also mentioned as contributing mechanisms. Treatment is generally conservative, but surgery may be required in severe cases. This paper presents the case of a 16-year-old girl with COVID-19-associated SPM who was treated conservatively in our department. The clinical course was favorable with resolution of respiratory symptoms and radiological (chest CT scan) image of pneumomediastinum. The patient was discharged 7 days after the confirmation of the initial SP diagnosis with appropriate treatment and recommendations for isolation. The sudden occurrence of chest pain and dyspnea should raise the suspicion of SPM in COVID-19 patients. Close surveillance and proper radiological monitoring are required in such cases. Treatment should be strictly individualized based on clinical course and radiological appearance.

## 1. Introduction

The pandemic generated by the SARS-CoV-2 virus throughout the world was a difficult test for the medical system, including the Romanian one. Many patients, especially adults, experienced severe respiratory symptoms and complications, including atypical pneumonia, acute respiratory distress syndrome (ARDS), and respiratory compromise, which required ICU admissions and mechanical ventilation. A small number of patients developed several types of rare respiratory manifestations, including spontaneous pneumomediastinum (SPM) [1,2].

Pneumomediastinum is defined by the presence of air in the mediastinal space. It can be spontaneous or secondary. Spontaneous pneumomediastinum (SPM) is a rare condition (1 in 33,000 hospital admissions or 1 in 12,000 pediatric presentations to the emergency department) [3]. It is defined by the presence of air in the mediastinum not related to trauma or iatrogenic procedure, such as endotracheal intubation or assisted mechanical ventilation. The etiology is considered to be diffuse alveolar damage and air leak through alveolar ruptures into the surrounding broncho vascular sheath-Macklin phenomenon [1,2,3,4]. Less commonly, pneumomediastinum can occur from air coming out of the upper respiratory tract, the intrathoracic airways, or the gastrointestinal tract secondary to increased intraluminal pressure or disturbed wall integrity [5]. Air brakes up at the hilum and diffuses into the mediastinum or through lax mediastinal fascia, reaching the subcutaneous tissues of the chest, neck, and upper extremities.

SPM is particularly common in newborns. There is a second peak of incidence during late infancy and early childhood, related to the high prevalence of respiratory infections in this age group [6]. The pathological mechanism is probably represented by increased pressure within obstructed airways. Lung necrosis secondary to viral infection could also be involved. There is a third peak during adolescence; tall, thin males are disproportionally affected, as is the case for spontaneous pneumothorax [7,8]. In this age group, Valsalva maneuver or coughing in adolescents with multiple possible triggers (asthma, laryngitis, exercise, and drug inhalation) was presumed to represent the likely mechanism of occurrence. Pneumothoraxes or pneumoperitoneum may also occur at the same time. Once the air enters the mediastinum, it can cause subcutaneous emphysema (Hamman syndrome). The association with respiratory infections could be reinforced by increased pressure within obstructed airways or by tissue necrosis from parenchymal infection [9].

SPM associated with SARS-CoV-2 infection was rarely reported in the literature. Most of the cases occur in adult patients; there are only a few cases of SPM diagnosed in children. The most likely mechanism of occurrence is probably barotrauma associated with cough in a patient who already has developed alveolar injury secondary to SARS-CoV-2 infection.

We present the case of a 16-year-old girl with SARS-CoV-2 infection treated in our hospital who was diagnosed with SPM after admission. The clinical course was favorable under conservative treatment. She was discharged with no respiratory symptoms and resolution of the SPNM changes on the chest CT examination.

## 2. Case Report

A 16-year-old girl presented to the hospital for ageusia and anosmia lasting for 3–4 days. Her previous medical history was significant for chronic gastritis and hypocalcemia; she had no allergies and was receiving no medication at that point. Given the epidemiological context, she was tested for SARS-CoV-2 infection and was found positive. She had no dyspnea, and the chest X-ray was not significant for SARS-CoV-2 changes (Figure 1a).

The clinical examination revealed a calm patient with no respiratory distress. Her height was 165 cm and she weighed 55 kg, with a BMI (Body Mass Index) of 20.2. The neck was supple on palpation, the oropharynx had no erythema, and the lungs were clear on auscultation. Her lungs were clear on auscultation; her respiratory rate was 14/min, and her heart rate was 68/min. Her blood pressure was 115/62 mm Hg. Peripheral saturation in O_2_ (SpO_2_) was 98% in room air. She was sent home in isolation and received treatment with Azithromycin, Dexamethasone, and analgesics according to COVID-19 protocol in use for mild cases. She returned after 4 days because of moderate dyspnea and chest pain; however, her general condition was good, and she required no O_2_ support. Her SpO_2_ in room air was 97%, her lungs were clear on auscultation, and she was hemodynamically stable (pulse rate 70/min, respiratory rate 16/min, blood pressure 112/65 mm Hg). Her labs at admission showed normal values for total WBC (white blood cell count), respectively, 6.98 × 10^3^/µL, neutrophils (3.75 × 10^3^/µL, 53.7% total WBC) and lymphocytes (2.26 × 10^3^/µL, 32.4% total WBC) with slightly increased monocytes (0.83 × 10^3^/µL, normal 0.15–0.70; 11. 9%, normal 3–7%). The mean platelet volume (MPV) was decreased (6.1 fL, normal 8–15 fL), but the platelet count was normal—304 × 10^3^/µL. Blood gases at admission: pH 7.38, pCO_2_ 43 mm Hg, pO_2_ 31 mm Hg, Lac 2.5 mmol/L (normal 0.9–1.9 mmol/L). Na^+^ 141 mEq/L, K^+^ 3. 9 mEq/L, Ca^++^ 1. 14 mmol/L. Blood chemistry was normal.

She was admitted for surveillance and started on Azithromycin, Plaquenil, IV hydration, cortisone, and analgesics. Analgesics are often prescribed in such cases, as there is no treatment scheme unanimously adopted by the medical staff [10]. WBC and blood chemistry ordered after admission showed decreased eosinophil count (0.04 × 10^3^/µL, normal 0. 05–0.7 × 10^3^/µL), increased creatin-kinase MB 2.7 U/L, normal < 25 U/L), increased lactatdehydrogenase LDH (493 U/L, normal 130–250 U/L), and increased D-dimers (9.04 µg/mL FEU, normal < 0.5 µg/mL FEU). C-reactive protein (CRP), T-troponin levels, and usual blood chemistry were normal.

The next day she complained of increased chest pain and a moderate degree of dyspnea, which raised the suspicion of a COVID-19-related respiratory complication, including pneumothorax or pneumomediastinum. Since the previous chest X-ray showed no abnormalities, a chest CT scan was recommended in order to identify such findings [10,11]. It showed no direct signs of pulmonary thromboembolism, no pulmonary condensation, and no pleural effusion. A moderate-sized pneumomediastinum (PM) was identified (Figure 1b). A pediatric surgery consult was required. Given the small size of the PM and the good general and respiratory condition of the patient, the recommendations included bed rest, symptomatic treatment, and clinical/radiological observation. Over the admission, her clinical status was good, with intermittent need for O_2_ delivered by nasal cannula and SpO_2_ of 96–98% in room air. She required no positive-pressure ventilation or ICU admission. Her vitals were stable at all times with clear lungs on auscultation. A second CT scan was performed after 9 days and showed resolution of the pneumomediastinum (Figure 1c). Additionally, she tested negative for SARS-CoV-2 the same day.

She was discharged after 10 days of hospitalization with a good health condition. She had no respiratory distress. The recommendations included 3-day isolation, vitamin supplements, and symptomatic treatment. At the 1-month regular follow-up, she was doing fine and had no respiratory complaints.

## 3. Discussion

Pneumomediastinum (PMS) has been rarely reported in patients with COVID-19, and most cases occur in adult patients. Isolated cases in adults were mentioned in Wuhan hospitals in early 2020, followed by other reports generally limited to a small number of cases. It was hypothesized that alveolar damage secondary to severe SARS-CoV-2 infection may represent the source of pneumothorax and pneumomediastinum in these patients. Barotrauma associated with cough in a patient who already has developed alveolar damage secondary to SARS-CoV-2 infection may precipitate the occurrence of pneumomediastinum. Mechanical ventilation and subsequent barotrauma were also related to the occurrence of pneumomediastinum in intubated patients.

Lopez Vega et al. mentioned that cough, which is a frequent symptom in COVID-19 patients, is the cause of pneumothorax and pneumomediastinum (P and P) [12]. They presented three cases of spontaneous P and P in ICU patients who did not require assisted mechanical ventilation. In their opinion, the mechanism of injury leading to P and P was represented by alveolar damage secondary to viral infection and rupture of alveolar wall due to increased pressure from pronounced coughing. Loffi et al. reported six cases of SPM in 102 COVID-19 patients investigated with chest CT (6% incidence) [13]. They associated the occurrence of SPM with severe cases of COVID-19 pneumonia. Three of their six cases had diffuse COVID pneumonia and spontaneous PMS, with later development of sudden ARDS that required aggressive management and intubation.

The occurrence of SPM associated with COVID-19 infection in a previously well child is not well documented. As of March 2021, there were only three cases reported in the literature [14]: in the first case, P and P were attributed to a transbronchial biopsy; in the second case, a 9-year-old child developed COVID-19 and pneumomediastinum secondary to a craniectomy procedure. In this patient, high ventilatory pressure used during surgery could have contributed to barotrauma. In the third case, a 17-year-old adolescent boy developed SPM. The mechanisms proposed were possible lung inflammation and 2 weeks of coughing resulting in barotrauma. In one case presented by Dixit et al., there were no previous cough or other symptoms, although widespread lung inflammation was present on imaging and LDH and ferritin levels were raised, which is typical of severe cases of COVID-19 [15]. Another study [16] reported pneumothorax in 2% of COVD-19-related pneumonia in children, and a few case reports in the literature have described those findings, usually in adolescents with severe disease.

Clinical presentation: according to several studies [14,15,16], the most common symptoms and their frequency are represented by dyspnea (40%), cough (32%), neck pain (17%), odynophagia (14%), and dysphagia (10%). The chest pain is located behind the sternum and has a pleuritic feature (with exacerbation during deep inspiration). The pain may radiate to the neck, shoulders, and arms. Other common complaints include light-headedness and weakness, and some patients present with neck swelling, torticollis, dysphonia, abdominal pain (typically epigastric), or back pain [16]. Low-grade fever may occur a few hours after the onset of the other symptoms.

Diagnosis of PMS is generally established by chest X-ray, with chest CT being performed only in complicated cases. Other radiological studies, such as ultrasound, are rarely performed but can help the diagnosis in children [17,18]. Chest CT representative findings in COVID-19 are represented by bilateral ground-glass opacities with or without consolidation involving posterior and peripheral lungs. Crazy paving pattern and reverse halo sign have also been described [19,20]. With further analysis of COVID-19 cases, a wide spectrum of CT imaging features was identified in these patients. CT changes can represent the only findings in asymptomatic or paucisymptomatic patients with COVID-19, including adolescents [21].

Clinical management: in most cases, movement of air through planes of subcutaneous tissues relieves the pressure within the mediastinum, allowing a conservative approach. The majority of patients require hospitalization for O_2_ delivery and observation, with up to 25% requiring ICU admission. Most patients show spontaneous resolution of PMS with conservative treatment (bed rest, analgesia, and O_2_ therapy) [22,23].

SPM is usually not accompanied by significant abnormalities of hematological parameters [22]. Similarly, most of the lab values were normal in our patient. Significant changes were recorded for D-dimer, troponin, CK, and LDH. Several studies have highlighted that high levels of LDH are significant for an increased risk of developing SPM [24].

The lack of data in pediatric age, together with the absence of specific management guidelines, lead to controversial approaches for the management of the condition in this population, currently, mainly on protocols established for adults. According to current data, it is very likely that this condition will become more and more frequent, representing, an adverse effect of the infection with COVID-19, being found in patients who have recovered from the disease [25].

We present the case of a 16-year-old girl with a mild form of COVID-19 infection who developed spontaneous pneumomediastinum (SPM) along the clinical course. She was admitted for chest pain and dyspnea, which suggested the occurrence of a respiratory complication associated with COVID-19 infection. The diagnosis was confirmed by chest CT scan. She responded well to conservative treatment and did not require surgery. As mentioned in several papers, the occurrence of SPM in COVID-19 cases can be explained by repeated or intensive coughing, sneezing, or vomiting episodes with acute increase of intrathoracic pressure [21]. Our patient did not relate such episodes. Another accepted theory is that alveolar inflammatory damage secondary to SARS-CoV-2 infection is responsible for alveolar septa rupture and further occurrence of SPM [26], which could be the case in our patient. This theory is supported by evidence that shows that diffuse interstitial alveolar inflammation infiltrates as the main histopathological finding in COVID-19 patients [27].

## 4. Conclusions

Spontaneous pneumomediastinum (SPM) is a rare complication of COVID-19, with few cases reported in children and adolescents. The sudden onset of chest pain and dyspnea should raise the suspicion of SPM in these patients. Increased levels of LDH may also suggest widespread alveolar damage with an increased risk of developing SPM. CT findings are very helpful for the purpose of diagnosis, especially in mild cases. Close surveillance and proper radio-logical monitoring are required in such cases. Treatment should be strictly individualized based on the clinical course and radiological appearance.

## Figures and Tables

**Figure 1 children-09-01641-f001:**
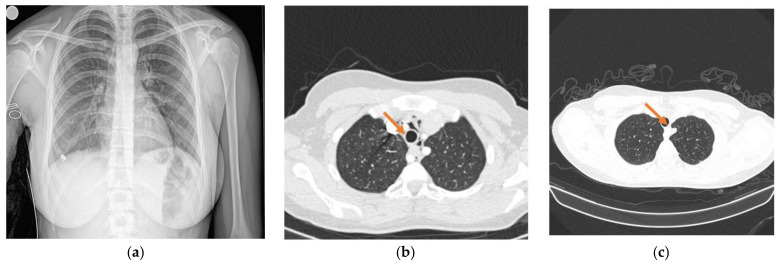
X-ray and CT images. (**a**) Chest radiograph—no abnormalities (Day 0); (**b**) chest CT—presence of pneumomediastinum (Day 4); (**c**) chest CT—pneumomediastinum resolution (Day 11).

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
