# Peer review of "Spontaneous Pneumomediastinum in a 16-Year-Old Patient with SARS-CoV-2 Infection: A North-East Romanian Case"

_children, 2022, doi:10.3390/children9111641_

Round 1

Reviewer 1 Report

The paper concerns spontaneous pneumomediastinum – a relatively rare complication in adolescents with mild COVID-19. Thus, I have read it with great interest. The paper requires some corrections, both content-related and editorial.

1.       I miss information on what the Authors consider as the cause of pneumomediastinum in the described case - what was the most likely pathomechanism? I would also appreciate information about the patient’s posture – her height and BMI. Was it of possible importance for the pneumomediastinum?

2.       The literature should be supplemented. To my knowledge, there are two other case studies about pneumomediastinum in adolescents that should be included in the Discussion:

·         Buonsenso D, Gatto A, Graglia B, Rivetti S, Ferretti S, Paradiso FV, Chiaretti A. Early spontaneous pneumothorax, pneumomediastinum and pneumorrhachis in an adolescent with SARS-CoV-2 infection. Eur Rev Med Pharmacol Sci. 2021 Jun;25(12):4413-4417.

·         Bellini D, Lichtner M, Vicini S, Rengo M, Ambrogi C, Carbone I. Spontaneous pneumomediastinum as the only CT finding in an asymptomatic adolescent positive for COVID-19. BJR Case Rep. 2020 May 15;6(3):20200051.

3.       The Authors should consider whether the discussion on pneumomediastinum in infants and young children is necessary. I would appreciate a more detailed discussion on adolescents instead.

4.       The Conclusions section contains the case description only – this should be included in the Discussion. Differences and similarities between the presented case and cases from the literature should be discussed.  The conclusions, however, should include the authors' recommendations regarding diagnostic and therapeutic approaches in such cases.

5.       Please note that the name of the pathogen causing coronavirus-2 disease is SARS-CoV-2 and the name of the disease itself is COVID-19. Thus one can write “COVID-19” or “SARS-CoV-2 infection” but not “COVID-19 infection”

6.       Rough English should be improved and careful proofreading by a native speaker is recommended.

7.       The Authors should carefully read the manuscript again and improve spelling mistakes like “biotrauma” (page 4, line 119) or “in 1 case, P and P were attributed” (page 4, line 129) – what does “P and P” mean? 

Author Response

23.10.2022

MDPI AG, St. Alban-Anlage 66

4052 Basel, Switzerland

Tel.: +41 61 683 77 34

Journal: Children

Special Issue: Lung Diseases in Children: From Rarer to Commonest

Re:         Manuscript ID children-1978489

Esteemed Reviewer:

We greatly appreciate the opportunity to have our work reviewed by you. The following phrases include the referee comments and our responses.

  1. I miss information on what the Authors consider as the cause of pneumomediastinum in the described case - what was the most likely pathomechanism? I would also appreciate information about the patient’s posture – her height and BMI. Was it of possible importance for the pneumomediastinum?

Response: Thank you for your review. We added this information on lines 62-68 and 82-85 (document with track-changes).

  1. The literature should be supplemented. To my knowledge, there are two other case studies about pneumomediastinum in adolescents that should be included in the Discussion:

  • Buonsenso D, Gatto A, Graglia B, Rivetti S, Ferretti S, Paradiso FV, Chiaretti A. Early spontaneous pneumothorax, pneumomediastinum and pneumorrhachis in an adolescent with SARS-CoV-2 infection. Eur Rev Med Pharmacol Sci. 2021 Jun;25(12):4413-4417.

  • Bellini D, Lichtner M, Vicini S, Rengo M, Ambrogi C, Carbone I. Spontaneous pneumomediastinum as the only CT finding in an asymptomatic adolescent positive for COVID-19. BJR Case Rep. 2020 May 15;6(3):20200051.

Response: Thank you. We appreciate your effort and time. We studied the specialized literature and supplemented with the references indicated by you. We added this information on 21 and 25.

  1. The Authors should consider whether the discussion on pneumomediastinum in infants and young children is necessary. I would appreciate a more detailed discussion on adolescents instead.

Response: Thank you. We added information on lines 168-170 and lines 181-189.

  1. The Conclusions section contains the case description only – this should be included in the Discussion. Differences and similarities between the presented case and cases from the literature should be discussed. The conclusions, however, should include the authors' recommendations regarding diagnostic and therapeutic approaches in such cases.

Response: Thank you. We added information on discussion part and we changed the conclusions (lines 224-229).

  1. Please note that the name of the pathogen causing coronavirus-2 disease is SARS-CoV-2 and the name of the disease itself is COVID-19. Thus one can write “COVID-19” or “SARS-CoV-2 infection” but not “COVID-19 infection”.

Response: Thank you. We modified. Please excuse us for overlooking this aspect.

  1. Rough English should be improved and careful proofreading by a native speaker is recommended.

Response: We corrected it and we hope that now everything is fine.

  1. The Authors should carefully read the manuscript again and improve spelling mistakes like “biotrauma” (page 4, line 119) or “in 1 case, P and P were attributed” (page 4, line 129) – what does “P and P” mean?

Response: We corrected it. “P and P” mean pneumothorax and pneumomediastinum (P and P) (lines 143).

Best Regards,

Roxana Gheorghita, Ph.D.

Corresponding Author
roxana.puscaselu@usm.ro

Reviewer 2 Report

-I would recommend improving table 1. It is disorganized and does not represent information relevant for SPM; while it represents more common COVID-19 infection biochemical findings.

- The Chest Xray at Day 0 is normal but it doesn't explain if there was any change, if there was any change in subsequent Chest Xray it would be worth adding, if not there is no need to have it included. 

- Well written article and precise discussion

Author Response

23.10.2022

MDPI AG, St. Alban-Anlage 66

4052 Basel, Switzerland

Tel.: +41 61 683 77 34

Journal: Children

Special Issue: Lung Diseases in Children: From Rarer to Commonest

Re:         Manuscript ID children-1978489

Esteemed Reviewer:

We greatly appreciate the opportunity to have our work reviewed by you. The following phrases include the referee comments and our responses.

  1. I would recommend improving table 1. It is disorganized and does not represent information relevant for SPM; while it represents more common COVID-19 infection biochemical findings.

Response: Thank you for comments. We appreciate your effort and time. We have given up table 1, but we have described the results from the laboratory in lines 93-99.

  1. The Chest Xray at Day 0 is normal but it doesn't explain if there was any change, if there was any change in subsequent Chest Xray it would be worth adding, if not there is no need to have it included.

Response: Thank you. With your permission, we decided to keep the image.

  1. Well written article and precise discussion.

Response: Thank you for your appreciation and comments. We also appreciate your effort and the time you gave to our manuscript.

Best Regards,

Roxana Gheorghita, Ph.D.

Corresponding Author
roxana.puscaselu@usm.ro
